# Research Progress on the Species and Diversity of Ants and Their Three Tropisms

**DOI:** 10.3390/insects14110892

**Published:** 2023-11-18

**Authors:** Hejie Dong, Xinyi Huang, Qingqing Gao, Sihan Li, Shanglin Yang, Fajun Chen

**Affiliations:** 1College of Plant Protection, Nanjing Agricultural University, Nanjing 210095, China; 12121228@stu.njau.edu.cn (H.D.); 12121226@stu.njau.edu.cn (X.H.); 12121225@stu.njau.edu.cn (Q.G.); 2022802167@stu.njau.edu.cn (S.Y.); 2College of Life Science, Nanjing Agricultural University, Nanjing 210095, China; 9211010109@stu.njau.edu.cn

**Keywords:** ants, species and diversity, phototaxis, chromotaxis, chemotaxis, green prevention and control, protection and conservation

## Abstract

**Simple Summary:**

Ants fulfill an important ecological function, not only serving as one of the main decomposers but also playing a role in improving soil fertility. Ants may conflict with humans as a pest species in homes and living facilities (e.g., closets, wardrobes, cabinets, and tables), as well as in outdoor facilities (e.g., tables, chairs, and benches). In addition, ants and humans can come into conflict in green spaces (including parks, gardens, and tourist attractions) and can seriously interfere with outdoor leisure and the life and entertainment activities of people (especially children). Therefore, it is a matter of public health and environmental protection to ensure ant prevention by adopting green control technology and not merely by spraying pesticides to kill them. This review mainly focuses on the species and diversity of ants, especially in China, and the research progress of ant tropism regarding phototaxis, chromotaxis, and chemotaxis (i.e., 3-tropisms). Moreover, current research on repellent substances is also summarized, analyzed, and discussed in order to help the application of green prevention and control technology for ant diversity protection and conservation, especially in daily home life and entertainment activities. This field of research is related to ensuring the healthy life of the public and the protection of the ecological environment.

**Abstract:**

Ants are one of the largest insect groups, with the most species and individuals in the world, and they have an important ecological function. Ants are not only an important part of the food chains but are also one of the main decomposers on the Earth; they can also improve soil fertility, etc. However, some species of ants are harmful to human beings, which leads to people’s panic or worry about coming into contact with these insects during their daily home life or in their tourism or leisure activities. The presence of ants in indoor living facilities and in outdoor green spaces, parks, gardens, and tourist attractions seriously interferes with the leisure life and entertainment activities of all people (especially children). How can we control ants in these environments? Do we kill them by spraying insecticides, or do we adopt green prevention and control technology for the ecological management of ants? This topic is related to healthy life for the public and the protection of the ecological environment. In this paper, the species and diversity of ants are introduced, and research progress regarding ant tropism is introduced according to the three aspects of phototaxis, chromotaxis, and chemotaxis (i.e., “3-tropisms”). The research on repellent substances from plants and insects and the related ant attractants are also summarized, analyzed, and discussed, in order to help the research and application of green prevention and control technology for ant diversity protection and conservation.

## 1. Introduction

Ants are one of the most diverse and abundant animal groups in the world, and they can be found everywhere, with more than 10,000 species [1]. Among them, many species are harmful to people (such as *Solenopsis invicta*, *Monomorium pharaonis*, etc.), and they are listed as high-priority quarantine pests by many countries [2]. These harmful ants will cause serious consequences after stinging people, resulting in people’s panic or worry about coming into contact with these ants in their daily home life or during outdoor tourism activities.

Ants may conflict with humans as a pest species in homes and living facilities (such as in closets, wardrobes, cabinets, and tables, etc.), as well as outdoor facilities (such as tables, chairs, benches, etc.). In addition, ants and humans can come into conflict in green spaces (including parks, gardens, and tourist attractions) and they can seriously interfere with the public leisure and entertainment activities of people (especially children). Ants can also damage agricultural equipment [3]. In addition, the problem of fire ants, leafcutter ants, termites, and other insects is very serious in tropical areas, causing extensive damage to crops and reforestation [4]. However, ants have an important ecological function; they are an important part of the food chain in ecosystems, not only serving as one of the main decomposers on the Earth but also playing a role in improving soil fertility. In nature, the ecological function of ants is equivalent to that of cleaners, shouldering the important task of purifying the environment, and they are also a food source and natural enemies for many animals. In addition to feeding on seeds, these ants can also, therefore, accidentally disperse them [5]. Also, they perform important ecosystem services such as pest control [6]. Therefore, it can be seen that the niches that they occupy are too important to be replaced by other organisms.

Even though ants are frequently encountered in people’s daily home life or in tourism and leisure activities, they must not be arbitrarily killed; harmless green management technology should be adopted to carry out the comprehensive prevention and control of ant colonies in order to protect diversity and for conservation purposes in these environments. Ecological pest control using insect tropism is considered to be an effective way to achieve green ecological pest control. Tropism is the directional behavioral response of insects to external stimuli. Among them, those who move in the direction of the stimulus are called positive taxis (i.e., seduction), and those who move away from the direction of the stimulus are negative taxis (i.e., avoidance). According to the nature of the stimulus sources, insect taxis mainly include phototaxis, chemotaxis, chromotaxis, thermotaxis, and humidity taxis, among which phototaxis, chemotaxis, and chromotaxis (namely, the “3-tropisms”) are widely used in pest control, monitoring, and early warnings [7,8,9,10,11,12]. At present, inductive killing and repellent technologies have been widely used in the monitoring and control of disease-carrying insects (such as mosquitoes, flies, cockroaches, fleas, etc.) [13]. In order to better carry out the comprehensive green and harmless prevention and control of ants in daily home life or during tourism and leisure activities, the diversity of ant species and their phototaxis, chromatism, and chemotaxis (the “3-tropisms”) have been reviewed and analyzed in this paper.

## 2. Ant Species and Diversity in China

Ants belong to the Hymenoptera group of insects in the family Formicidae. They have many ecological functions, such as pollinating plants, controlling pests, improving soil fertility, and maintaining ecological balance etc. At present, a total of 14,084 species of ants have been found, belonging to 347 genera, and with 16 subfamilies [1]. There are more than 800 species of ants known in China, the most common ones being the small yellow house ant (also known as the Pharaoh ant), big-headed ant, sword-jawed smelly house ant, smelly ant, black ant, etc. According to the published domestic literature in China [14,15,16,17,18,19,20], there are a total of 863 species of ants known in China, totaling 7 subfamilies and 90 genera (see Appendix A).

Several studies on ant species diversity have been carried out in China. Li et al. (2011) investigated the populations and species diversity of ants in four types of vegetation on Zijin Mountain and found that the dominant species were *Camponotus japonicus* (24.80%), *Pheidole nodus* (17.98%), *Pristomyrmex pungens* (17.57%), and *Paratrechina flavipes* (14.31%) [21]. Han et al. (2021) studied ant species diversity in the eastern Daliang Mountains and found that there were 6 subfamilies, 43 genera, and 136 species of ants. They indicated that altitude was the main factor affecting the ant community indexes of species number, individual density, and community diversity, and these indexes generally decreased with an increase in altitude [22]. Yang et al. (2022) found that the dominant species of ants were *Technomyrmex albipes*, *Tapinoma melanocephalum*, and *Carebara affinis*; simultaneously, they discovered three new species recorded in China, i.e., *Kartidris matertera*, *Pheidole planifrons*, and *Pseudolasius zamrood* [23]. Most species of ants are non-toxic or harmless, but a few species are poisonous and even aggressive, causing pain and itching, severe shock, and even death, such as the more widely distributed red fire ant *S. invicta*, the army ant, red ant, small yellow house ant (also known as the Pharaoh ant) *M. pharaonis*, the double-toothed thorn ant *Polyrhachis dives*, the big-toothed savage ant *Odontomachus bauri*, the Argentine ant *Linepithema humile*, and the native harvest ant *Messor barburus*. Reared as pets, the latter is known as the most aggressive or venomous species of ant. It was reported that the little fire ant (LFA), *Wasmannia auropunctata*, successfully invaded mainland China in 2022; an invasion of LFA will harm the health of human beings and animals, cause the persecution of local biodiversity, and harm agricultural production, and represents an important pest in invasive areas [24]. In addition, the common groups of ants in buildings include *M. pharaonis*, carpenter ants, *Pheidole rhombinoda*, *Tetramorium caespitum*, *Iridomyrmex anceps*, *Polyrhachis vicina*, etc.; they can cause damage to buildings, contaminate food, and spread diseases, among other effects. Among them, *M. pharaonis* generally chooses the place that is closest to food and water sources in the building, nests in the cracks of the building, and invades the rooms to bite human bodies, causing red spots, pain, and itching; they especially like to bite a baby’s delicate skin, causing disease. In view of the species-specific particularity and ecological service functions of ants, alongside killing some of the toxic and dangerous species that are harmful to human health, a protective attitude should be adopted for most groups of ants. Even if it is necessary to control ants in special environments (such as homes, leisure and entertainment places, public green spaces, etc.), green and harmless methods of comprehensive prevention and control measures should be adopted for ant diversity protection and conservation.

## 3. Phototaxis in Ants

The light sensed by insects is mostly short-wave light that is near the broad-spectrum center of the electromagnetic wave, with a wavelength range of 253–700 nm. This is equivalent to the section from the ultraviolet light range in the spectrum to the inner part of the infrared light range [25]. Insects can both recognize color and see short light waves beyond the human eye. Phototaxis is the tendency response to the induction of a specific range spectrum by the photoreceptor cells in the visual organ, while spectral sensitivity describes the relationship between the response threshold and the wavelength of the insect compound eye or photoreceptor to light stimulation, i.e., the proportion of absorption of incident light and the generation of electrical signals. To date, there are several hypotheses about insect phototaxis, which can be listed as follows: (1) the optical orientation behavior hypothesis (OoBH): Many nocturnal insects use a celestial body as a reference for their phototaxis, with their body’s vertical axis perpendicular to the celestial body and horizontal with the insect’s body line. Night-time lights are also used as directional references by insects, but this reference is much closer than the referenced celestial body. As a result, the insects spiral toward the light, which eventually causes the insects to approach the light source [26]. (2) Biological antenna hypothesis (BAH): This posits that insect photosynthesis is caused by courtship behavior and that the various protrusions, depressions, and thread structures on insect antennae are similar to contemporary antenna devices. The insect’s antennae can sense the pheromone’s molecular vibration, while the far-infrared spectra in the attractive insect lights are consistent with the vibrational lines in the pheromone molecules; therefore, the insects react to this information, leading to phototaxis [27]. (3) Light interference hypothesis (LIH): The hypothesis suggests that the dazzling effect on nocturnal insects in dark zones interferes with their normal behavior. Due to the low brightness of the dark zone, insects are unable to return to the dark area to continue their activities, resulting in an attraction to the light lamp [7].

The light source is the main factor in attracting and gathering insects. The previous literature has indicated that the wavelength, intensity, and polarization of a light source can affect the phototactic behavior of insects [9,11,12]. In general, the spectral reaction-sensitive areas of insects are mainly concentrated in the ultraviolet light (350 nm), blue light (440 nm), and yellow-green light (540 nm) spectra. Each species of insect varies in terms of their specific wavelength preference within these three segments [9,28]. Within a certain range of light intensity, insect phototactic behavior gradually increases as the light intensity increases. Wang et al. (2016) found that the phototaxis index of *Rhyzopertha dominica* at 20,000 and 10,000 lx light intensities was significantly higher than that at 5000, 2500, and 1500 lx light intensities [29]. Moreover, different types of polarized light differ in their ability to attract insects. Zhang et al. (2021) indicated that light intensity had no significant effect on the phototaxis rate of adult oriental armyworms (*Mythimna separata*), while the phototactic behavior of armyworms of *M. separate* was mainly affected by the light wavelength; the sensitive light wavelengths of female and male adults were also different [30]. Gong and Liu (2011) found that *Drosophila* larvae display light avoidance behavior [31]. According to the distribution of absorption peaks, the photoreceptors of insects are divided into three categories, i.e., short-, medium-, and long-wavelength types [32]. The spectral sensitivity of small-eye receptors in the dorsal margin varies according to insect species; bees, ants, and flies are sensitive to ultraviolet light, crickets and locusts is sensitive to blue light, and Coleopteran beetles are sensitive to green light. Insect behavioral responses to UV light are on two levels, namely, UV sensitivity and UV vision [33]. UV sensitivity is the ability of insects to sense and detect UV light, i.e., they have photoreceptor cells in the retina that absorb UV light and can convert these optical signals into electrical signals [34]. Many insects have UV sensitivity and can sense a certain UV light wavelength being reflected by light sources or objects in their surrounding environment, causing them to perform corresponding trend or avoidance reactions [35,36]. At present, there are few reports on the study of ant phototaxis, although it is generally believed that ants exhibit no phototaxis or have only a limited phototaxis response to short, medium, and long light waves. There is still a lack of sufficient experimental evidence and research on this aspect needs to be strengthened urgently. However, light stimulation experiments to determine the phototactic behavior of ants at specific wavelengths show that ants have a clear preference for UV light [37]. Based on the three main insect photoreceptors (i.e., UV, green, and blue light sensitivity), ant taxa are more sensitive to purple and green, or there are species specificity differences that require further experimental verification [9,28,38].

Light-trap killing technology is a commonly used technique for pest control that is based on insect phototaxis, and its development in China has gone through five stages [39]. Firstly, in the 1950s, incandescent lamps, oil lamps, steam lamps, and other ordinary lights were used to light-trap and kill insect pests, although the controlling efficiency of such lights in terms of killing pests is limited [40]. Secondly, during the 1960s and 1970s, black lights and high-pressure mercury lamps were developed, and the controlling efficiency of such developed lights in terms of killing pests was significantly improved. However, their light spectrum was broad, with poor selectivity regarding the induced insect species, high potential safety risks, and high costs [41]. Thirdly, in the 1990s, the frequency-vibration-type trap-killing lamp was developed, which overcomes the shortcomings of incandescent lamps, black lamps, high-pressure mercury lamps, etc., by using different wavelengths of light sources in combination with color and odor to trap specifically targeted insect pests [39]. Of course, using a high-voltage power grid to kill insects presents potential dangers to human and animal security. Fourthly, in the early 21st century, a new light source (i.e., LED) trap lamp was developed; this kind of lamp has high brightness, low energy consumption, and long usage life. A single-wavelength LED light source can also solve the problem of poor selectivity regarding target insects, due to its wide spectral range; it can also improve the safety level and enhance the trap-killing efficiency regarding the targeted insect pests. Finally, recently, a new intelligent trap-killing lamp system based on LED and other trap lamps integrates automatic counting, image recognition, and network transmission for a green approach to the prevention and control of ants.

## 4. Chromotaxis in Ants

In the long duration of their evolution, some insects have developed specific-species color tropism (i.e., chromotaxis), a tendency seen in response to the photoreceptor cells in their visual organs to induce light waves, which is also known as “color vision” [8]. As early as the 1950s, chromotaxis was used to control insect pests in Europe, America, and Japan. In the 1960s, yellow plates were used as traps for aphid control, and chromotaxis was also applied for insect pest management in Chinese Taiwan in the 1980s [10]. As shown by the visual ecology technology represented by insect retinal potential determination technology (i.e., the electroretinogram or ERG), the chromotaxis of insects refers to the tendency of their visual organs regarding the induction of light and color, which is reflected in the positive attraction to and negative avoidance of a light or color source [10]. Thus, insect chromotaxis is a form of phototaxis in nature. The chromotaxis of insects to color can be divided into positive and negative tropisms, while the tendency toward color is positive, and that of avoiding color is negative [8]. Different species of insects may have species-specific choices and inclinations regarding chromotaxis, and there are also differences in color tropism between females and males or among the different developmental stages for the same species of insects [10].

Most insects have color vision, but there are differences in their color vision between/among different species of insects, depending on the spectral sensitivity and the interaction of the involved photoreceptors [32]. Like many other animals, insects use color information during the day and at night to guide their behavior, e.g., for locating their specific habitats, identifying some conspecifics, and orientating themselves according to celestial and terrestrial landmarks [32]. In insects, color vision involves different physiological and neuronal processing stages that mainly require a comparison of the output of at least two types of photoreceptor (PR) cells within the retina [42].

Ants, like most other Hymenoptera, have eight PR cells on each small ocular membrane, with a long striated muscle and a short striated muscle forming a central striated muscle. Some early histological studies utilized the radial movement of the PR cells during selective pigment adaptation to determine the different spectral receptor types and their arrangement within the small ocular membrane. These studies were based on the phenomenon that the location of pigments in PR cells depends on the intensity and spectral composition of the incident light, indicating that there are at least two PR types in ants that are sensitive to UV and green light, respectively [38]. There was also an electrophysiology study based on retinal mapping, which indicated that ants might have retained the ancestral “trichromy” seen in wasps, while more species of ants may have lost their blue light receptors during evolution [43]. It was found that ants preferred beads with less light transmission and also preferred green beads, from which two hypotheses of intensity discrimination and color discrimination were presumed through the experimental study of color perception and ant preference [44].

In the practice of agricultural production, researchers and technicians developing plant protection use the chromotaxis of insects to make colored sticky plates to trap insect pests that damage crops. For example, aphids, whiteflies, leafhoppers, leaf miners, and other insect pests show strong chromotaxis to yellow colors, meaning that yellow sticky plates are made to control these insect pests; thrips show strong chromotaxis to blue colors, and so blue sticky plates are made to control this type of insect pest [10]. In addition, according to the negative tendency of aphids regarding silver and gray colors, silver and gray plastic film, aluminum foil, black plastic film, or spraying white powder emulsion to crop plants can play a role in avoiding aphids. At present, color-based trapping technology using colored sticky plates is one of the main promoted green prevention and control technologies used for insect pest management in agricultural production [8,10].

## 5. Studies of Ant Chemotaxis Based on Avoidant and Attractant Substances

Insect chemotaxis refers to the insect’s behavioral response to chemical stimuli in the surrounding biological and abiotic environment through its olfactory organs, manifesting as positive chemotaxis (i.e., attraction effects) and negative chemotaxis (i.e., avoidance effects). The use of chemotaxis for pest control is a traditional technology developed in the 1960s, which has the advantages of high sensitivity and strong specificity, allowing it to achieve good control efficiency when combined with physical and chemical measures, and it is also widely used in monitoring the density of disease vectors [13]. Studies on the chemotactic responses of ants have also focused on their attractant and avoidant behavior responses. Among them, the repellent substances of ants are mainly divided into two aspects, i.e., plant- and insect-derived repellent substances (Table 1).

### 5.1. Plant-Derived Repellent Substances for Use against Ants

Plant-derived repellent substances mainly include turpentine, the natural components of cinnamon, linalool repellent, acacia volatiles, volatiles from ant-repellent flowers, eucalyptus secretions, tea tree oil, etc. (Table 1). Turpentine is mainly extracted from the tree resins of the Pinaceae family of plants, using the distillation method, and its main component is terpene. A variety of terpene-based ant repellents can be isolated from turpentine as raw materials, all of which can be synthesized via α-pinene or β-pinene; among the 17 isolated terpene-based ant repellents, 11 of them have a repellent rate of more than 80% against ants, while nopropyl acetate even reaches a 100% repellent effect against ants [45]. The sesquiterpenes elemol and β-eudesmol, found in the essential oil from eucalyptus leaves, have a certain repellent effect on *Atta sexdens rubropilosa* [46]. Wang et al. (2014) found that the feeding and attack rates of *S. invicta* after fumigation with eucalyptus and mugwort oils were significantly lower than in the control, so these two plant-derived oils could be used as alternatives to commercial repellents and fumigants used against the red imported fire ant [47]. The camphor tree is typically found in China and can produce largely secondary defensive substances during its growth; the main chemical component of camphor, being a natural pest control substance, encourages strong evasive activity in many insects (including ants). Another mainly secondary control substance is camphor essential oil, and its main component is a terpenoid compound of safrole with strong repellent activity against insects (including ants). These two secondary products of camphor and its essential oil can be combined to make a camphor ant repellent, which has very good repellent activity against ants (such as *M. pharaonis*) and not only triggers a touch response effect but also repels ants [48]. It was found that the untreated control group was moved toward the nest by workers of the leaf-cutting ants (*Acromyrmex ambiguus* and *A. lobicornis*), while oatmeal soaked with 1% tea tree oil was moved in the opposite direction to the nest in experiments using oatmeal soaked with tea tree oil; it was speculated that tea tree oil contains plant volatiles that have a repellent effect on ants and the repellent effect on ants increases with an increase in the concentration of tea tree oil [49]. There is a high content of cinnamyl acetate in cinnamon leaf essential oil and its insecticidal effect is stronger than that of bark essential oil. When the essential oil contained trans-cinnamaldehyde and cinnamyl acetate with a 2:1 ratio, the red imported fire ant *S. invicta* was affected due to the mixture interfering with antenna morphology and recognition receptors [50]. Cheng et al. (2008) indicated that both the indigenous cinnamon leaf essential oil and *trans*-cinnamaldehyde had an excellent inhibitory effect in controlling *S. invicta* [51]. Moreover, the repellents of cinnamon essential oil and p-anisaldehyde showed high avoidance efficiency against six species of ants, including *Cardiocondyla batesii*, *Plagiolepis pygmaea*, *P. schmitzii*, *Solenopsis* sp., *T. nigerrimum*, and *T. semilaeve*, where the pipes of a subsurface drip irrigation (SDI) system containing the two selected compounds remained almost intact, with no damage being caused by the ants [3].

However, although acting as flower-visiting insects, ants are rarely beneficial to plants. During flower-visiting, ants always tend to disrupt pollination by blocking other flower-visiting insects or by stealing nectar, which causes some plants to develop ant-repellent properties [61,62]. For example, acacia plants secrete volatile organic compounds (VOCs) such as linalool, geranyl, α-pinene, and limonene, to reduce ant flower visits (i.e., the evocative effect) and also reduce nectar loss, thus ensuring the smooth progress of pollination [52]. Another example is the male flowers of *Petasites fragrans*, which secrete 4-methoxy-benzaldehyde, a compound that has a markedly repellent effect on ants and also significantly reduces the flower-visiting duration of ants [53].

### 5.2. Insect-Derived Repellent Substances for Use against Ants

Some insects can secrete and release certain compounds (e.g., VOCs) to discourage ants. Studies have shown that ants, especially in tropical and subtropical regions, prey on wasp larvae, which is not conducive to the reproduction of the wasp population. In response to this long-term selective pressure, many species of wasps secrete special repellent substances to repel ant invasion. For example, Brazilian wasps (e.g., *Solenopsis geminata*, *Forelius pruinosus*, and *Pheidole* sp.) can secrete a defensive pheromone, the main component of which is methyl palmitate, to avoid ant invasion [54]. As another example, the females of some wasp populations can avoid the invasion of ants by rubbing their abdomen and leaving ant-repellent substances at the edges of ant nests; the main components of the released repellent substances are unsaturated acids (e.g., palmitoleic acid, linoleic acid, and oleic acid) [55]. The thrips *Suocerathrips linguis* secretes a low-volatile defensive substance to avoid ants, and the active ingredient in this secretion is (11Z) -11, 19-docodiene acetate, which can effectively discourage 85–90% of ants [4]. In addition, the glandular secretions of wasp subfamily individuals also have repellent effects on ants. Since the main component of the alarm pheromones secreted by many aphids is β-farnesene (especially (cis-) β-farnesene and α-farnesene), it demonstrates a certain repellent activity [56,57]. All the above insect secretions can be used in the development of ant repellents to achieve the green and sustainable control of ants.

### 5.3. Other Sources of Repellent Substances for Use against Ants

Besides these plant- and insect-derived repellent substances for use against ants, there are also repellent substances from other sources that can be used to avoid ants. Bridge-ring Terpenoids are important terpene-repellent substances for ants, and seven bridge-ring terpenoid compounds (including nopol, nopyl methyl ether, nopyl ethyl ether, nopyl n-propyl ether, nopyl formate, nopyl acetate and nopyl propanoate)have demonstrated high repellent activity against ants [58]. A series of citronellal acetals generated after acetalation, using citronellal as raw materials, have shown high repellent activity for *M. pharaonis*, and both citronellal diethyl acetal and citronellal 1,3-malondial acetal have a more than 85.33% and 97.10% repellent rate against *M. pharaonis* at 2.5–10 mg/mL, respectively [59]. In addition, the food additives ethyl anthranilate and butyl anthranilate, even at extremely low concentrations (<100 µL/L), showed high repellent efficiency against nesting by the workers of fire ants, *S. invicta*; therefore, these chemicals should be evaluated for further potential applications in preventing the spread of fire ants [2]. Diatomite soil is an important mineral formed when the remains of single-celled algae diatoms are deposited in oceans or lakes after death, and it is used in the control of insect pests. It has been shown that 85% diatomite powder can kill ants (*Blattella germanica* and *M. pharaonis*) by destroying the waxy layer of their skin and resulting in water loss, which has a good anti-ant effect. When used as an inert powder, it will not pollute the environment, is non-toxic to mammals, and can be used in combination with repellents to better control ants [60]. In addition, there are some very useful common substances to repel ants that can be used effectively for daily ant prevention, such as the odor volatiles of coriander, celery, leeks, walnut leaves, tobacco, and peppercorns, along with lemongrass essential oil, lavender essential oil and perfume, charcoal ash and rubber bands from burning firewood, etc., which have shown better repelling efficiency against ants.

### 5.4. Ant Attractants

Ants have a propensity to seek out sweet foods and animal materials (especially insect carcasses); thus, these materials can be used as ant attractants for their prevention. Buehlmann et al. (2014) showed that the compounds that guide the desert ant, *Cataglyphis forage*, in locating food are fatty acids that are widely present in insect cuticles, and their attraction is triggered by the decomposition of fatty acids. The response of ants to a series of compounds released by dead insects was tested in this experiment, of which 3 (i.e., indol, (E)-2-octenal, and linoleic acid) out of 15 tested odors attracted about half of the tested individuals. Linoleic acid turned out to be the most attractive odorant tested; therefore, it could act as an effective attractant for ant prevention [63]. Linoleic acid is found in animal fats, existing in the form of glycerides along with other fatty acids, which are also found in insect cuticles [64]. For example, the linoleic acid percentage in beef and pork is 1.8% and 6% respectively, which shows that the smell of pork can attract ants more effectively than the smell of beef. In addition, Buehlmann et al. (2014) tested changes in the ability of dead insects to attract ants according to distance and found that the number of responding ants decreased with an increase in the distance from food. The results showed that 50% of the ants had a reaction distance of 3.3 m, and the maximum reaction distance was 5.9 m [63]. In addition, Gu and Wei (2021) studied the chemotaxis of ants toward sugars and sweeteners; they concluded that when sweeteners and sugars are present at the same time, ants have very low or no chemotaxis to sweeteners, while they have high chemotaxis to fructose, and the order of ant tropism was fructose > sucrose > maltose > glucose. The chemotaxis of ants toward glycosides was very low or almost no chemotaxis, and only the chemotaxis of ants toward xylitol was relatively high [65]. In addition, in the range of 0.04–0.12 g/mL, the chemotaxis of ants toward sucrose increased with an increase in concentration, but there was no response difference between the different concentrations of fructose solution [65]. Therefore, higher concentrations of fructose as well as sucrose solutions can be used as good attractants for trapping ants.

## 6. Discussion and Future Prospects

In this paper, a list of the various ant species and their diversity was generated and summarized, based on the related literature produced in China (see Appendix A). The related research progress regarding the “3-tropisms” of ants (comprising phototaxis, chromotaxis, and chemotaxis) was discussed and systemically summarized in order to guide research into and the future manufacture of more environmentally friendly, green, and harmless methods of prevention and control against ants in household and field environments (Figure 1). At present, there are many documents on ant chemotaxis, both at home and abroad, while there are few reports on ant phototaxis and chromotaxis. Although it is generally believed that ants have no phototaxis or that they have a certain phototactic response to different wavelengths of light, there is a lack of sufficient experimental evidence to support these hypotheses. It has been confirmed that many species of ants have evolved to lose their blue-light receptors [43], meaning that they exhibit a sensitivity response to both ultraviolet and green light [41], especially to ultraviolet light [40]. Research in this field needs to be further strengthened in the future.

In addition, this paper focuses on the attraction and avoidance responses seen in ant behavior; the existing research on ant chemotaxis was systematically reviewed in terms of the ant repellents found in plant- and insect-derived substances and from other sources, as well as ant attractants, in order to provide data support for the research and development of ant chemotaxis-based attractants and repellents and to enable the ecological protection, green prevention, and control of ants in household and field environments. In general, the main measure to control ants involves spraying chemical pesticides in the ants’ gathering areas. Ant taxa form an important group with ecological and service functions in the environment; ants are not only an important link in the food chain but also act as one of the main decomposers on the Earth, and they also play a key role in improving soil quality and fertility. Therefore, the prevention and control of ants should not be achieved by spraying pesticides to kill them; some green or harmless methods of prevention and control should be carried out for the ecological management of ants, which is related to the healthy life of the public and for ecological environmental protection. Therefore, ant control should focus on the development of green prevention and control technologies that are based on phototaxis (e.g., light-trapping and repellents), chromotaxis (e.g., color-plate trapping and repellents), and chemotaxis (e.g., attractants and repellents) to achieve effective prevention and the sustainable protection of ant populations in the household and field environments.

## Figures and Tables

**Figure 1 insects-14-00892-f001:**
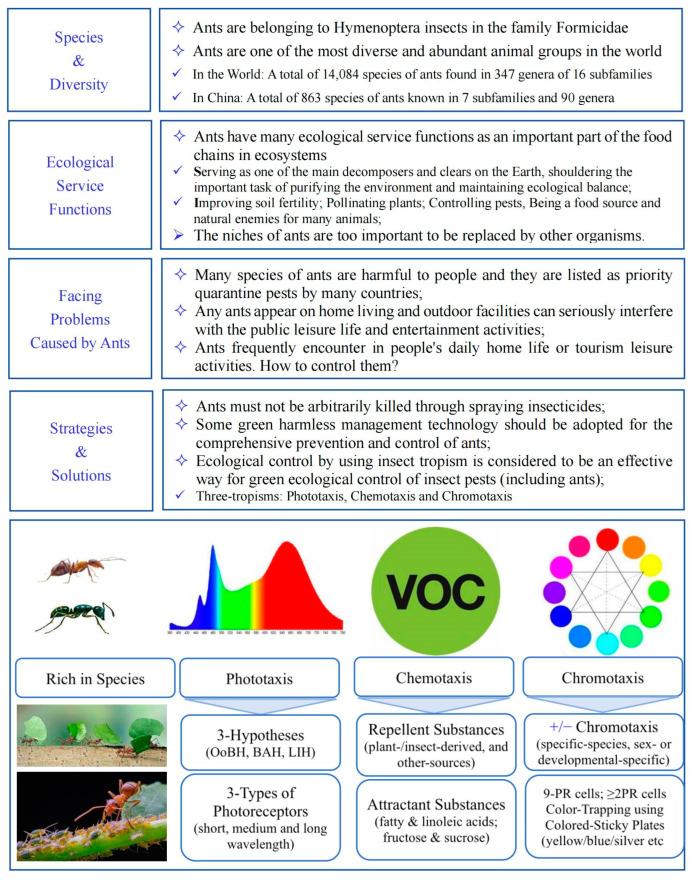
The various species and diversity of ants and their “3-tropisms”, reviewed and analyzed in order to better carry out comprehensive green, harmless prevention and control of ant groups in daily home life and during tourism and leisure activities in green spaces. (Note: VOC—volatile organic compound; OoBH—optical orientation behavior hypothesis; BAH—biological antenna hypothesis; LIH—light interference hypothesis; +/− denotes a positive/negative response).

**Table 1 insects-14-00892-t001:** Repellent substances derived from plants and insects for use against ants.

Type of Repellents	Species of Plant/Insect/Other	Main Substances	Target Ants	References
Plant sources	Pine tree	Turpentine (mainly terpenes, turpentine terpenoids, novartis bligh acetate, etc.)	*Monomorium pharaonis*	[45]
	Eucalyptus	Leaf essential oil (mainly sesquiterpenol and beta eudinol)	*Atta sexdens rubropilosa*, *Solenopsis invicta*	[46,47]
	Camphor tree	Camphor (mainly camphorone) and Camphor essential oil (mainly safrole)	*M. pharaonis*	[48]
	Tea tree	Tea tree oil (repellent plant volatiles)	*Acromyrmex ambiguous* and *A. lobicornis*	[49]
	Cinnamon	Leaf and bark essential oils (mainly cinnamyl acetate and *trans*-cinnamaldehyde)	*S. invicta*; *Cardiocondyla batesii*, *Plagiolepis pygmaea*, *P. schmitzii*, *Solenopsis* sp., *Tapinoma nigerrimum Tetramorium semilaeve*; *S. invicta*	[3,50,51]
	Acacia	Plant volatiles (mainly linalool, geraniol, α-pinene and limonene)	*Crematogaster sjostedti*, *C. mimosae*, *C. nigriceps*, or *Tetraponera penzigi*	[52]
	Anise	p-anisaldehyde	*C. batesii*, *P. pygmaea*, *P. schmitzii*, *Solenopsis* sp., *T. nigerrimum*, *T. semilaeve*	[3]
	Mugwort	Plant essential oils	*S. invicta*	[47]
	*Petasites fragrans*	Floral volatiles (mainly 4-methoxybenzaldehyde)	*Formica aquilonia*	[53]
Insect sources	*Polistes fuscatus*	Defensive pheromone (mainly methyl palmitate)	*Solenopsis geminata*, *Forelius pruinosus*, and *Pheidole* sp.	[54]
	*Polistes dominulus* and *P. sulcifer*	Abdominal secretion of female bees (mainly unsaturated fatty acids such as palmitoleic acid, linoleic acid, and oleic acid)	*Crematogaster scutellaris*, *Formica cunicularia*, and *Lasius* sp.	[55]
	The pointy thrips	Defense substance ((11Z)-11,19 eicosadienyl acetate)	*Myrmica rubra*	[4]
	Aphids	β-farnesene (especially (cis-) -β-farnesene and α-farnesene)	Ants	[56,57]
Other sources	Terpenoids	Bridge-ring Terpenoids (nopol, nopyl methyl ether, nopyl ethyl ether, nopyl n-propyl ether, nopyl formate, nopyl acetate and nopyl propanoate)	*M. pharaonis*	[58]
	Citronellal	Citronellal acetals (citronellal ethyl diacetal, citronellal 1, 3-malondialdehyde)	*M. pharaonis*	[59]
	Food additives	Ethyl anthranilate and butyl anthranilate	*S. invicta*	[2]
	Diatomite soil	Diatomite powder	*Blattella germanica*, *M. pharaonis*	[60]

## Data Availability

Not applicable.

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
