# Peer review of "Research Progress on the Species and Diversity of Ants and Their Three Tropisms"

_insects, 2023, doi:10.3390/insects14110892_

Round 1
Reviewer 1 Report
Comments and Suggestions for Authors
Thanks for this submission - which is a good overall review of relatively non-toxic (green methods) that might be used to deter ants in human public spaces (a laudable goal). Your paper specifically attempts to group these interventions into methods using the '3 tropisms' (phototaxis, chemotaxis, and chromotaxis). This is a somewhat novel approach - given a somewhat extensive literature on ant management. Overall, I was happy with your paper, but I would recommend some English language editing. Overall, The English writing is serviceable and comprehensible, but some moderate editing would be helpful.
For example, the following sentence from MS lines 47 - 52 could be edited for clarity in the English writing: 'Any ants appear on home living facilities (such as closets, wardrobes, cabinets, tables, etc.), as well as outdoor facilities (such as tables, chairs, benches, etc.) servicing for people resting and playing in green spaces (including parks, gardens and tourist attractions), that can seriously interfere with the public leisure life and entertainment activities of people (especially children).' This sentence might be rephrased in this way: 'Ants may conflict with humans as a pest species in home living facilities (such as closets, wardrobes, cabinets, and tables, etc..), as well as outdoor facilities (such as tables, chairs, benches, etc.). In addition, ants and humans can come into conflict in green spaces (including parks, gardens, and tourist attractions) and can seriously interfere with the public leisure life and entertainment activities of people (especially children).'
This is just one example. While the English is intelligible as-is (and the overall message of your paper is clear) some moderate editing might improve the readability of this article.
Comments on the Quality of English LanguagePlease see my previous suggestions for English-language editing...
Author Response
Response: Here, many thanks for your evaluation on the topics of our manuscript. And thanks again for your comments on some English language editing.
We have revised the sentence from the primary MS lines 47 - 52 based on your suggestion for clarity in the English writing. And we also invited Dr. Sabin Saurav Pokharel helped revising and editing all the English language of the version (insects-2660410R1). Please check the version of insects-2660410R1 as the attachment.
Thanks again for your help revising our manuscript!

Reviewer 2 Report
Comments and Suggestions for Authors
This manuscript holds highly significant information related to ants diversity, their economic importance and environment friendly approaches for their conservation and management. The Manuscript has been formatted exact focusing on the title, but I have some suggestions for the improvement of the manuscript in the attached file. As a general comment, please add diagrammatic illustrations to each section of the manuscript.

Miner editing required
Author Response
Here, many thanks for your valuable suggestions, which will be helpful to improve our review manuscript. We have revised based on your comments, which are following as:
Abstract:
Q1. Line 9-12 please make this long sentence into two or more small sentences for meaningful understanding of the reader.
Response: We have made the deletion as you suggested, and now the position in the text is 32-35 lines.
Q2. Page 2 please this whole page may be somewhere adjusted in the main text for illustrations.
Response: We have placed the picture behind Discussion and prospects.
Introduction
Q1. Line 1-2 need proper citation.
Response: We have cited the appropriate sources in the original text, and the number cited is [1], and now the position in the text is 54-55 lines.
Q2. Page 3 first paragraph (42-56) and second paragraph (58-82), in both you are first describing role and economic importance of ants then coming to control. Please make both as a single one and follow the same format in you followed in first paragraph or in second any of them.
Response: We have made the deletion as you suggested, and now the position in the text is 60-88 lines.
Q3. Page 4 line 105-125 you have started to discuss about harms of ants in Australia and Sahara region etc. But the heading of this part of manuscript is Ant species and diversity in China. Please you must only discuss ant’s diversity and species of ants from China only according to the heading.
Response: We have modified it as you suggested, and now the position in the text is 125-136 lines.
Q4. Line 128-129 need proper citations.
Response: We have cited the appropriate sources in the original text, and the number cited is [26], and now the position in the text is 150-152 lines.
Q5. Line 153 change Some documents to previous literature.
Response: We have modified it as you suggested, and now the position in the text is 173 lines.
Phototaxis in ants
Q1. Line 128-176 Although this part is very useful and informative but must be shortened to focus the main theme Phototaxis in ants.
Response: We have made the deletion as you suggested, and now the position in the text is 158-203 lines.
Q2. Line 186-205 need proper citations due to the presence of actual facts.
Response: We have modified it as you suggested, and now the position in the text is 204-222lines.
Chromotaxis in ants
Q1. Line 210-215 need proper citations.
Response: We have cited the appropriate sources in the original text, and the number cited is [8] and [10], and now the position in the text is 224-237 lines.
Q2. Line 230-231 need citation.
Response: We have cited the appropriate sources in the original text, and the number cited is [33], and now the position in the text is 238-240 lines.
Q3. Line 232 -236 it is a very long sentence and may lose concentration of the reader. Make small 2-3 sentences.
Response: We have made the deletion as you suggested, and now the position in the text is 252-255 lines.
Discussion and prospects
Q1. Line 382 (seen in Table S1?????????????????????/.
Response: Table S1 is the attached table uploaded to the editorial department. And here, we also added this table (Table S1) on the end of the Manuscript. Please check this in the version (insects-2660410R1+Table S1).
General Comments
The authors have made a significant contribution in the field of Myrmecology and addressed the social issues due to ants in varied ecological conditions. They have proposed environment options for their management and conservation in our ecosystem. Every section of the manuscript needs photographic illustration.
Response: Here, many thanks for your evaluation on the topics of our manuscript. We add the pictures of the Graphical Abstract (GA) for the photographic illustration of the manuscript. Please check this change in the Word file (insects-2660410R1).
